# Measurement of the Mapping between Intracranial EEG and fMRI Recordings in the Human Brain

**DOI:** 10.3390/bioengineering11030224

**Published:** 2024-02-27

**Authors:** David W Carmichael, Serge Vulliemoz, Teresa Murta, Umair Chaudhary, Suejen Perani, Roman Rodionov, Maria Joao Rosa, Karl J Friston, Louis Lemieux

**Affiliations:** 1Biomedical Engineering Department, School of Biomedical Engineering and Imaging Sciences, King’s College London, London SE1 7EH, UK; david.carmichael@kcl.ac.uk; 2Developmental Imaging and Biophysics section, UCL Great Ormond Street Institute of Child Health, London WC1N 1EH, UK; suejen.perani@skope.ch; 3Department of Clinical and Experimental Epilepsy, UCL Institute of Neurology, Queen Square, London WC1E 6BT, UK; serge.vulliemoz@hcuge.ch (S.V.); umair.chaudhary@ucl.ac.uk (U.C.); r.rodionov@ucl.ac.uk (R.R.); 4Epilepsy Unit, Neurology Department, University Hospital and University of Geneva, 1211 Geneva 14, Switzerland; 5Department of Bioengineering, Institute for Systems and Robotics, Instituto Superior Tecnico, Universidade de Lisboa, 1049-001 Lisbon, Portugal; 6Department of Computer Science, University College London, London WC1E 6BT, UK; mariajoao@gmail.com; 7Wellcome Trust Centre for Human Neuroimaging, UCL Institute of Neurology, Queen Square, London WC1N 3BG, UK; k.friston@ucl.ac.uk

**Keywords:** functional MRI, Intracranial EEG, BOLD, BOLD coupling, fMRI biophysics, EEG-fMRI

## Abstract

There are considerable gaps in our understanding of the relationship between human brain activity measured at different temporal and spatial scales. Here, electrocorticography (ECoG) measures were used to predict functional MRI changes in the sensorimotor cortex in two brain states: at rest and during motor performance. The specificity of this relationship to spatial co-localisation of the two signals was also investigated. We acquired simultaneous ECoG-fMRI in the sensorimotor cortex of three patients with epilepsy. During motor activity, high gamma power was the only frequency band where the electrophysiological response was co-localised with fMRI measures across all subjects. The best model of fMRI changes across states was its principal components, a parsimonious description of the entire ECoG spectrogram. This model performed much better than any others that were based either on the classical frequency bands or on summary measures of cross-spectral changes. The region-specific fMRI signal is reflected in spatially and spectrally distributed EEG activity.

## 1. Introduction

Functional magnetic resonance imaging (fMRI) has emerged as the pre-eminent neuroimaging modality for studying functional segregation—and increasingly integration [1]—reflecting its capacity to map distributed hemodynamic (blood oxygen level dependent or ‘BOLD’ signal) changes at the scale of millimetres over the entire human brain. Although BOLD signal changes are considered a marker of underlying neuronal activity, the neuronal basis of the fMRI signal remains the subject of intense investigation. This is largely because of its importance for understanding, modelling and inferring underlying brain processes [2,3]. Animal studies have shown that BOLD signal changes are most closely related to local field potentials in the high gamma-band range [4] and reflect input and intra-cortical processing [3]. However, there are discrepancies regarding which frequency band [3,4,5] or combination of frequency bands [6,7,8] is the best correlate of the BOLD signal, which could be task-, species- or brain system-dependent [9]. There is further uncertainty about how BOLD responses relate to electrophysiological activity occurring at different frequencies, with both BOLD decreases and increases (relative to a task or state baseline) reported [10,11]. Previous studies have largely focused on particular frequencies or classical EEG frequency bands to predict BOLD changes; however, state-related EEG changes are often cross-spectral [12,13]. There have been reports of BOLD changes being well represented by the cross-spectral metrics of the overall EEG signal change such as the ‘root mean square frequency’ [13], however, these models have not been tested in data with a wide spectral range or in different states such as during rest.

Furthermore, the spatial aspect of this relationship is relatively neglected. For example, fluctuations in oscillatory activity, which may be related to information transfer and network control [14,15], recorded in one location, could index synaptic and metabolic changes in a remote region. Additionally, the spatial extent of electrophysiological activity (e.g., motor activity and motor inhibition) is likely to be distributed over several cortical areas [10,16].

Intracranial EEG recordings are performed for localisation purposes in some medically refractory patients with focal epilepsy who are candidates for epilepsy surgery; intracerebral depth electrodes that penetrate the brain (so-called ‘depth EEG’ or ‘stereo-EEG’: SEEG), provide exquisitely localised data, while subdural grid electrodes placed on the cortex (ECoG) capture activity over larger cortical regions. Importantly, these recordings do not attenuate high-frequency activity, unlike scalp EEG which suffers from inherent spatial-temporal filtering and muscle artefacts. While spatial sampling is limited by clinical considerations, icEEG has greater sensitivity and spatial specificity in relation to electrophysiological activity, compared to its scalp-based counterparts [17,18]. Finally, there is also more limited evidence of coupling between BOLD and electrophysiology in the resting state [19,20,21] although some work has been performed to evaluate average responses to natural stimulus tasks applied to each modality sequentially in humans [22,23,24].

In this study, we used simultaneous ECoG-fMRI acquired in three patients with epilepsy during invasive pre-surgical investigations. ECoG coverage of the sensorimotor cortex provided a unique opportunity to explore the coupling between electrophysiology and fMRI, with exquisite spatial precision and greater spectral range than available from scalp recordings, while accounting for spontaneous activity and variability both during task and rest conditions. These patients were selected based on the suitability of the implantation coverage for the present analysis from a pre-surgical series with varied implantation configurations, each specified by the clinical team based exclusively on considerations related to their value for providing localizing information for putative curative surgery. Using these data, we were able to characterise the relationship between EEG activity and fMRI signal changes within the sensorimotor cortex during a motor task and contrast it with the equivalent coupling in the resting state. We aimed to determine: (1) the best model of BOLD changes, in particular, comparing frequency-specific and cross-spectral EEG models of fMRI; (2) if the best model was state dependent, and (3) characterise the sensitivity of this relationship to the colocalisation between the EEG and fMRI signals.

## 2. Methods

Three patients with refractory focal epilepsy were studied based on the ECoG electrode coverage of the sensorimotor cortex that offered a good prospect of detecting effects related to a simple motor task. Following a comprehensive safety assessment to ensure accordance with safety guidelines and approval from our local ethics committee (Joint UCL/UCLH), the participants were recruited and written informed consent obtained. Each patient underwent a tailored ECoG implantation (Table 1) based on the hypothesis generated from long-term scalp video-EEG recordings and other clinical data.

A carefully devised experimental protocol was followed to ensure patient safety, based on testing of the exact experimental configuration used for scanning and a large margin of safety to account for possible differences between in vitro experiments and in vivo studies [25,26,27]. Following completion of the clinical ECoG recordings, cables connecting the electrodes to the amplifiers were replaced by customized 90 cm cables for simultaneous ECoG-fMRI recording, bundled, rerouted to the vertex of the head, re-bandaged and laid out precisely along the scanner’s central *z*-axis [27]. All implanted electrodes were recorded from when amplifier capacity allowed, otherwise, targeted electrodes were selected (Table 1).

### 2.1. Data Acquisition

We performed MRI with a 1.5 T Siemens TIM Avanto scanner (Siemens, Erlangen, Germany) with a head transmit-receive coil, low SAR scans (≤0.4 W/Kg head average), exact electrode cable placement along the scanner and RF coils’ *Z*-axis running towards the head end of the [27]. The following scans were performed (1) localiser, (2) FLASH T1-volume, TR/TE/flip angle = 150 ms/4.49 ms/25°, (3) 10 min gradient echo EPI (TR 3 s/TE 40 ms/flip angle 90°, 38 slices that were 2.5 mm thick with a 0.5 mm gap, 3 × 3 mm in-plane resolution) in the resting state where the patients were instructed to lie still with their eyes closed (4) 10 min gradient echo EPI where a visually cued opposing finger-to-thumb task was performed for 300 s with the same EPI parameters as above with 30 s blocks of left vs. right hand. In all subjects

ECoG signals were recorded using an MR-compatible amplifier system (128 channels) (Brain Products, Munich, Germany) during fMRI acquisitions. The EEG recording system sampling at 5000 Hz was synchronized to the scanner’s 20 kHz gradient clock to allow direct correlation over time. The system was located at the head end of the scanner and ECG was recorded using a 16 bipolar channel ExG MR compatible system located at the patient’s feet. Carbon fibre leads were used for the ECG recording in addition to spatial separation to prevent potential interactions between the intracranial electrodes and leads with the ECG; both were recorded using Brain Recorder (Brain Products, Gilching, Germany). Recordings sampled at 5000 Hz with subsequent filtering and down sampling to 250 Hz.

### 2.2. Data Processing

Scanning-related artefacts on EEG were removed using the Brain Analyser V1.3 (Brain Products, Munich, Germany) implementation of the template subtraction and filtering algorithm [28] and data was referenced to the average.

All EPI images were realigned to the first image and spatially smoothed (FWHM 8 mm). The presence of significant fMRI changes correlated with the effect of interest (see below) was assessed voxel by voxel over the whole field of view using SPM software (www.fil.ion.ucl.ac.uk/spm accessed on 15 October 2023) in Matlab (www.mathworks.com accessed on 15 October 2023).

For the five-minute finger tap task, a standard block design was used in a general linear model—with motion modelled as the six realignment parameters and a voxel-wise statistical threshold of *p* < 0.05 family wise error corrected for multiple comparisons across voxels was applied to the ensuing SPM of t statistics.

The position of each electrode contact was determined on a CT scan (obtained for clinical purposes) co-registered to the space of the EPI data. The distance from each electrode contact to a cluster of fMRI activation was calculated as the minimum Cartesian distance over all voxels in the cluster.

We sought to characterise the coupling between the fMRI signal in the hand sensorimotor region and spatially distributed ECoG activity as a function of frequency for two brain states: rest and motor task. This was achieved by calculating the correlation over time between the fMRI signal averaged over the task-activated hand sensorimotor region and the fluctuations in ECoG power as a function of frequency and ECoG electrode contact location, after convolution with a hemodynamic response function (HRF) [29] (Figure 1):

Spectral analyses were performed for each electrode contact on the gradient artefact corrected ECoG data using a Morlet wavelet transform [31] as implemented in SPM8 (www.fil.ion.ucl.ac.uk/spm accessed on 15 October 2023) with a wavelet factor value of 7, to obtain time-frequency data for each electrode at frequencies 1, 3, 5, …, 99 Hz, giving time-frequency spectrograms at 2 Hz spectral sampling.

The primary hand sensorimotor region was defined as the fMRI task activated region. The realigned (but not spatially smoothed) data from this region was extracted and averaged before having first order drifts in fMRI signal removed. The time-frequency signals described above at frequencies 1, 3, 5, …, 99 Hz were convolved with the standard ‘canonical’ HRF in SPM before being re-sampled at the time corresponding to the start of each scan. Local correlations were calculated between each ECoG-frequency model and the measured fMRI response from the hand sensorimotor region. This was performed for data obtained both during the task and periods of rest (as described above). A significance threshold of *p* < 0.001 was used, corresponding to a level of *p* < 0.05 Bonferroni corrected for multiple comparisons across frequencies. This represents a very conservative threshold because the EEG has significant correlations over frequencies (whereas Bonferroni correction assumes independence between each frequency). Given that the data is also spatially smooth (spatially non-independent) [32] and the initial threshold is very conservative we did not correct further for multiple comparisons across electrode contacts.

We aimed to determine the best EEG-based model of fMRI fluctuations, at rest and during the task. We compared a family of models, grouped as follows: firstly, single- versus multiple-predictor models to determine if the fMRI could be best modelled by a single EEG feature or was best modelled by a more complex representation. Secondly, we compared models based on classical EEG frequency bands between themselves, and to cross-spectral EEG measures.

For the model comparison for each channel EEG, the time-frequency data described above was (1) averaged based on frequency-bands (delta: 1, 3 Hz; theta: 5, 7 Hz; alpha: 9, 11 Hz; beta: 13, 15, …, 31; gamma Low: 33, 35, …, 51 Hz; gamma high: 53, 55, …, 99 Hz) before being convolved with the canonical hemodynamic response function (HRF) and re-sampled at the start of each scan volume (3 s temporal resolution).

Models were built with individual spectral band regressors (single predictors) and the power in all of the six chosen frequency-bands (multiple predictor model). Motion was modelled as the 6 realignment parameters and slow temporal drifts were modelled using a cosine filter or order 1.

We calculated each of the family of cross-spectral single predictor models described in the Appendix and compared them to the models based on classical frequency bands described above. Finally, we created a multiple predictor cross-spectral model using a parsimonious data driven model of the complete spectrogram by performing a Principal Component Analysis (PCA) and taking the components that described >90% of the variance. Typically, this yielded the most complex model with 10–18 repressors giving an indication of the rich spectral information content of the intracranial EEG data.

## 3. Model Comparison

Bayesian model comparison (as implemented in *spm_PEB.m*, spm8 [33]) was used to compare the different EEG-derived models’ ability to predict the BOLD signal changes. The model evidence (log evidence) for each of the EEG-derived models described above was calculated and the relative evidence compared to the null model (containing motion and slow temporal drifts). In this context, a value of 3 is equivalent to 5/100, and 20 to 1/1000 of the model being better by chance (i.e., equivalent to a *p*-value of 0.05 and 0.001, respectively in classical statistics). It should be noted that the models are penalised for complexity and therefore the evidence scores the model that explains the most variance given the complexity; therefore, allowing for a conservative comparison of models with increased complexity.

## 4. Results

### 4.1. Frequency-Specific ECoG-fMRI Correlation during a Motor Task vs. Resting State

First, we identified the electrode exhibiting the strongest correlation for each subject. The strongest correlation was always found at an electrode located either directly above the hand primary motor cortex or immediately posterior to the central sulcus (Figure 2a–c). Note that due to the spatial relationship between the ECoG grid at the surface and the cortical folding, broadly, this contact was always overlying the motor cortex. During the motor task, the maximum positive correlation as a function of frequency was found in the high gamma band for all three subjects (91 Hz for subject #1 and 69 Hz for subjects #2 and #3, Figure 2a–c, respectively, black) and the strongest negative correlation was found in the beta band (17 Hz for subject #1, 15 Hz for subject #2 and 29 Hz for subject #3). In the resting state, the profile of the coupling—as a function of frequency—was similar but the correlation was weaker and more variable between subjects (Figure 2a–c, red). The most distinctive feature of the coupling profile during rest (Figure 2a–c, black) was a significant negative correlation in the low beta range (<20 Hz).

In Figure 3a–c, the correlation for each subject was mapped spatially over the cortex for the significant positive peak in correlation (gamma frequency range) and negative peak in correlation (beta frequency range, see Figure 3d–f).

The dotted line represents the position of the central sulcus, and the numbers correspond to the electrodes numbering system (see Table 2); * indicates significant correlation *p* < 0.001 uncorrected.

Second, we investigated the coupling’s spatial dimension by mapping the fMRI-ECoG correlation across the ECoG grid. During the task (Figure 3) a region of very strong positive correlation at high gamma band frequencies was revealed at three post-central contacts located within 5 mm of the task-activated region (Figure 3a–c). In the beta range a strong negative correlation was seen over a much wider area of cortex including cortex that was more distant from the task-activated region in the pre- and post-central cortex (Figure 3d–f). During rest, the main feature was a significant (but weaker than during the task) negative correlation at frequencies in the alpha-beta range in all three subjects (Figure 4). During the motor task in the beta band the region of negative correlation extended over a wide cortical area, extending beyond the primary motor cortex.

### 4.2. Comparison of Frequency-Specific and Cross-Spectral EEG Models of fMRI

The individual classical frequency band-based fMRI predictors were beta, and high gamma power based on previous observations [7]. Two multiple-predictor models were designed to summarise the entire spectrogram: first, a previously used model comprising of the power in each of the classical frequency bands (delta, theta, alpha, beta, low gamma and high gamma) [6]; second, cross-spectral variation obtained by principal component analysis of spectrogram (the components explaining 90% of the EEG variance were used in the model, resulting in 10–18 regressors). All of these EEG predictors were convolved by the haemodynamic response function. Five cross-spectral summary metrics were designed to embody alternative aspects of the EEG spectrogram as a single predictor of BOLD changes. These consist of three spectral first order moments (‘means’): the root mean squared frequency (so-called ‘Kilner heuristic’) (‘*q_RMSF_*’) [13], a modified version of the latter that amplifies the influence of the dominant frequency (‘*q_MSF’_*’), the spectrogram centre of mass (*I_CofM_*); and two spectral second order moments: about its mean, and about 40 Hz (I40 Hz) based on the point of inflection for the correlation between fMRI signal and EEG in Figure 2. See Appendix B for definitions and the Appendix A for simulations illustrating the cross-spectral single predictor characteristics.

Model evidence is shown for each model and subject relative to a ‘null model’ containing only nuisance regressors (six realignment parameters and a high pass cosine filter). A value of 3 is considered strong evidence (equivalent to *p* < 0.05), we considered the results to be significant across individuals when the sum was >9 and each subject individually had a positive model evidence. These values are shown in bold.

These comparisons were performed for the ECoG channel showing the highest correlation in the previous analysis (which in all subjects was within 5 mm of the task fMRI activation) and across all ECoG grid channels (Figure 5) using empirical Bayesian model comparison [24].

#### 4.2.1. Motor Task

First, we wanted to determine the best EEG-derived predictor of BOLD in the primary sensorimotor hand area during the motor (finger-tap) task, for the ECoG channel near (<5 mm Euclidian distance) to the fMRI activation in the primary sensorimotor cortex. We compared the log of the ratio of model evidence of each model to a model containing only nuisance effects [33]. High gamma power was the best single-predictor electrophysiological measure of fMRI changes during the motor task. The PCA model performed similarly well, despite its increased complexity (which is strongly penalised by the model comparison method used) and was therefore both the best cross-spectral and multiple-predictor model.

We note that the multiple-predictor model based on classical frequency bands performed poorly compared to the other models, in particular, compared to the PCA model. This suggests that averaging within classical frequency bands destroyed significant predictive information.

Second, we examined the spatial distribution of model performance. The spatial maps of the log evidence for the best performing models in each category *q_MSF’_*, high beta- or gamma-band and PCA are shown in Figure 5. The high gamma band and *q_MSF’_* models are good predictors only for contacts that are highly co-localised, whereas the PCA model performs well from nearly all electrode contacts including in distal (hierarchically higher) brain areas. The beta band predictor was also a good model of BOLD changes over a wide region of cortex. This is consistent with the wider region of cortical inhibition during ipsilateral finger tapping. Taken together, our results strongly suggest that there is an electrographic representation of the task over a much wider region of cortex than that which shows strong fMRI changes—and that this representation is distributed in a spatially structured way over frequencies.

#### 4.2.2. Resting State

At rest, the pattern was more variable between subjects and model evidence was lower across the board (Table 2). The PCA model performed best and was the only metric to be better than the null model in all subjects. The best cross-spectral single predictor model was the *q_MSF’_* and the best individual frequency band model was beta band power. In patient #1, although the PCA model was the best, model evidence was low. This may be due to the ‘irritative zone’ (generating interictal epileptic activity between seizures) overlapping with the primary sensorimotor region and disrupting normal ongoing resting-state activity. In the remaining subjects, there was strong evidence for the beta-band and PCA models explaining the BOLD signal fluctuations from the fMRI defined sensorimotor cortex. The spatial distribution of model performance at rest (see Figure 5) showed that a PCA model performed well over the primary sensorimotor cortex in all subjects, as did beta frequency band power. In contrast to the task data, high-gamma power was not a good BOLD predictor most likely due to its relative absence in this state.

## 5. Discussion

We investigated the coupling between sensorimotor BOLD and electrophysiological signal variations in humans by mapping the correlation between the average fMRI signal from the hand sensorimotor region and frequency-specific ECoG signals. To do this, we used unique recordings of simultaneous electrophysiological signals on the surface of the human brain (ECoG) and whole-brain fMRI. In particular, the capability to record high-gamma activity over the motor cortex and fMRI together in humans enabled us to study the coupling between electrophysiological and hemodynamic fMRI signals as a function of brain state and localization. The intracranial EEG signal has been shown to be of high fidelity and able to record high frequency activity [34].

### 5.1. Local Coupling

In summary, we found task-related BOLD to reflect mainly high gamma-frequency power fluctuations in the local vicinity of the fMRI activated region. In contrast, at rest, beta power in spatially distributed contacts best predicted (inversely) BOLD signal changes. The best BOLD predictor based on a single cross-spectral metric was *q_MSF’_* which models fMRI changes as shifts in the mean of the square of the spectral power such as those that can result from power fluctuations at specific frequencies (see Appendix B for details on the cross-spectral metrics’ behaviour). This is consistent with the notion of at least partially independent neuronal sources of EEG changes at specific frequencies [35]. There were distinct peaks in the alpha-beta range with higher correlation (Figure 2), suggesting that there was a relationship between specific oscillatory activity and fMRI signal changes. In contrast, at high gamma frequencies, there was a high correlation across a broad frequency range. We found the PCA-derived model performed best across task and rest, and when considering the electrophysiological activity recorded over the entire ECoG grid. This model retains variance either from power changes at a given frequency or frequency shifts of any nature—it is simply a parsimonious description of the entire spectrogram. Taken together this may explain the PCA model’s large performance advantage over a model containing classical frequency band which have previously been suggested to be optimal [6].

### 5.2. Spatial Coupling Aspects

We found task-related BOLD to reflect mainly high gamma-frequency broad-band power fluctuations in the vicinity of the BOLD response while beta band activity over a much larger cortical area was inversely correlated with the fMRI activity.

At rest, beta power in spatially distributed contacts (inversely) predicted BOLD signal changes but there was not the same focal positively collocated gamma activity. The beta-band activity was prevalent during rest and during contralateral hand movements, inversely correlating with fMRI over a wide area of the sensorimotor cortex. This confirms it as a general feature when attending to or suppressing movement [36]. In contrast, gamma activity was highly localised in motor regions and only present during motor activity. It is possible that the relatively low power of the gamma activity made it more difficult to measure in the context of greater noise levels in ECoG recordings during fMRI although we have previously performed evaluation of this data [37].

### 5.3. Novelty of Our Electrophysiological BOLD Predictors

Some of our findings confirm previous reports: for example, gamma-band activity was strongly coupled to BOLD [4,5,38], but only during the task. Strong positive correlations in the gamma-band were found in conjunction with an inverse relationship at lower to mid frequency bands consistent with the idea that fMRI signal increases reflect shifts in EEG power from low to high frequencies [13]. These findings (Figure 2a–c) are consistent with studies in the auditory cortex using non-simultaneous fMRI (pre-operative) and ECoG in two patients with epilepsy [5].

In the resting state, the coupling was reduced, although it had a similar form to the frequency dependence seen during the task (Figure 2). Beta-band power was the most predictive frequency-specific predictor of BOLD changes during rest (and was inversely correlated), potentially indexing the relative level of cortical inhibition, consistent with a scalp EEG-fMRI study [7].

The best cross-spectral metric was *q_MSF’_* which allowed for fMRI changes both from shifts in power from low to high frequencies and increased power within a frequency in the absence of other changes at specific frequencies (see the Appendix A for the details of the cross-spectral metrics). This might be expected given that different neuronal populations are thought to be responsible for generating these signals [39].

We also evaluated multiple predictor models based on classical frequency bands and a cross-spectral model using PCA of the entire spectrogram. When considering all brain states and whether or not the ECoG channel was close to the fMRI region, we found the BOLD responses were best predicted by the PCA multi-spectral model. Because the PCA model is substantially more complex, it must explain far more BOLD variance than any other model to retain the greatest model evidence on marginal likelihood, given that our model comparison heavily penalises model complexity. Crucially, and in contrast to previous studies in the non-human primate visual cortex [6], our data does not support the multiple predictor model based on the classical frequency-bands as the best predictor of BOLD. This may be explained by the fact that our model comparison properly penalised model complexity—but could also be due to differences between brain regions and species.

Previous studies have found some variability in the neurophysiological features that best explain BOLD fMRI changes from both single spectral EEG frequencies, cross-spectral models [8] or models containing several frequency bands [6]. Crucially we have demonstrated that the results of these analyses are sensitive both to the subject’s state and to the spatial relationship between the two modalities.

Our results suggest that focal fMRI changes are accompanied by widespread EEG changes. The EEG changes predicting fMRI were not limited to specific frequencies or a close co-localisation; therefore, a cross-spectral PCA model is likely to be effective when looking at different brain regions that exhibit activity at different frequencies. From the reverse perspective, this suggests that fMRI changes can be linked to a plethora of EEG changes and cannot be in general interpreted as relating to a single specific EEG feature. The exception to this rule was found during the activity of primary cortex, where there was a close correspondence between high gamma power and fMRI, consistent with previous studies [3,4,5,38]. This is entirely consistent with predictive modelling of resting state connectivity in EEG and fMRI, where EEG connectivity in all frequency bands could predict fMRI to a greater extent than fMRI could predict EEG based connectivity [39].

The ability of ECoG-fMRI to demonstrate the link between fMRI fluctuations and (oscillatory) neural activity should help refine biophysical models, particularly in accounting for the emerging properties of neuronal populations (e.g., ref. [40]) and for subsequent application in pathology [41]. More generally, the characterisation of the coupling shown here, in different brain states, represents vital information for the interpretation of fMRI responses in terms of the underlying processing [1]. For example, it can be crucial for informing and validating efforts to model the underlying distributed brain responses such as the effective connectivity between neuronal populations [1]. There is increasing evidence of cortical layer specific spectral features of the EEG signal [35]. This would provide a biophysical explanation of the potential independent contributions to BOLD changes [7]. The strong predictive performance of the PCA model suggests that there are widespread cortical EEG changes that reflect focal BOLD signal changes and this complexity, if characterised, could provide important new information and avoid the misinterpretation of neurophysiological and fMRI comparisons.

### 5.4. Epilepsy and ECoG-fMRI: Feature for Study and Potential Confound

The patients studied suffer from epilepsy so inferring behaviour in the healthy population requires some caution. However, electrode placement strategies are designed to test contrasting localisation hypotheses (devised following the results of non-invasive tests) and frequently include electrodes placed within or over regions presumed to be unaffected to map the eloquent cortex. Intracranial EEG recordings in patients with epilepsy represent a unique opportunity to access a wide range of brain regions for study under conditions involving relatively normal physiology (e.g., ref. [5]).

The ECoG is exquisitely sensitive to local epileptic electrophysiological features, which should enable the effects of epilepsy to be accurately modelled and removed from studies aiming to make inferences about ‘normal’ brain function. This also provides an opportunity to map and understand fMRI changes associated with epileptiform events, including high frequency oscillations [41,42,43] with unprecedented sensitivity, because we have demonstrated the ability to utilise high frequency ECoG data obtained with simultaneous fMRI.

### 5.5. Further Limitations

Owing to the rarity of patients with highly similar implantations and the extremely challenging logistic constraints imposed by the limited amount of time that the icEEG electrodes are in situ, but not available to record clinically useful data, it is not easily possible to study large groups. Despite a level of reproducibility across our three subjects, we acknowledge that these numbers preclude high confidence in their generalisation.

While the relationship between BOLD fMRI and EEG is important to understand because it underlies the neurobiological interpretation of fMRI results it does not have clear clinical significance and might be altered in pathology.

## 6. Conclusions

During motor activity, high gamma power was the only frequency band where the electrophysiological response was co-localised with fMRI measures across all subjects. The best model of fMRI changes across states was its principal components, a parsimonious description of the entire ECoG spectrogram. This model performed much better than any others that were either based on the classical frequency bands or summary measures of cross-spectral changes. The region-specific fMRI signal is reflected in spatially and spectrally distributed EEG activity.

## Figures and Tables

**Figure 1 bioengineering-11-00224-f001:**
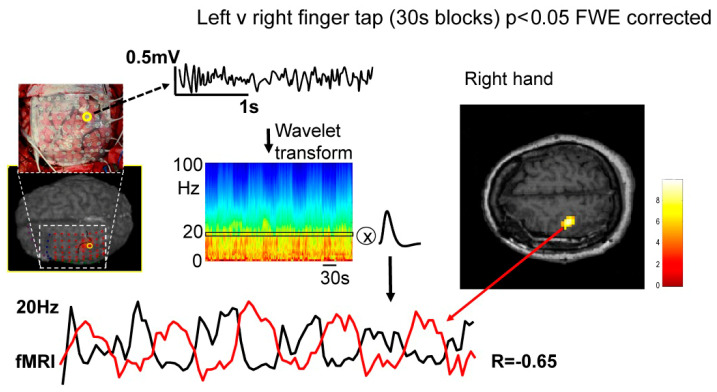
Analysis overview. The process of comparing fMRI to ECoG data is shown with ECoG data from the post-central ECoG (contact #22, subject #1 top left corner) transformed into time-frequency space via a wavelet transform before being convolved with the HRF. This yields a spectrally specific model of the fMRI changes (e.g., black line at the bottom) for correlation (or model comparison) with the fMRI signal in the hand sensorimotor region (red line at bottom; average from task activated region, *p* < 0.05 FWE corrected). This region was used due to the strong evidence that it is commonly active both during rest and task [30]. The correlation between ECoG power at 20 Hz and fMRI signal during the task is shown to be strong and negative. This correlation analysis was repeated for each frequency during the task and rest to establish spectral specificity (see Figure 2a–c) and then for each electrode contact to establish spatial specificity (see Figure 3 and Figure 4).

**Figure 2 bioengineering-11-00224-f002:**
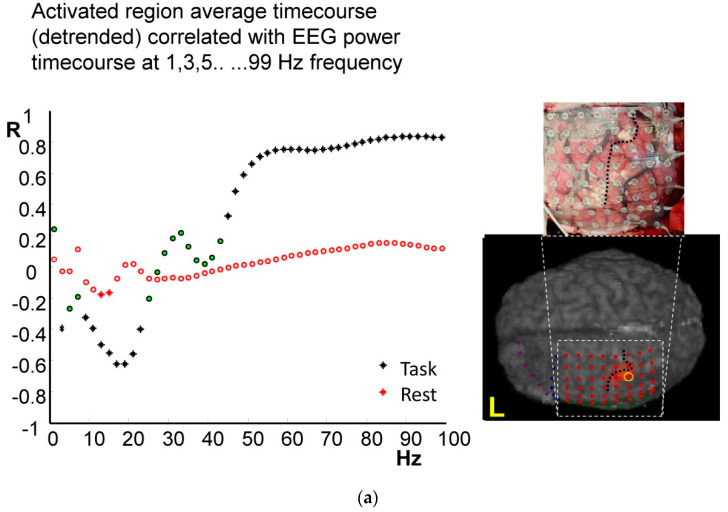
Frequency-specific sensorimotor cortex correlations between colocalised ECoG and fMRI during task and rest. For each subject (**a**–**c**) the fMRI signal from the fMRI task activation defined hand motor area was correlated with the co-localised ECoG data from a single electrode contact. The location of the electrode contact used is highlighted by a yellow circle on a reconstruction of the individual’s cortical surface and ECoG contact locations a photo is also provided where available (subjects #1 and #3). Stars indicate significant correlation *p* < 0.001 which corresponds to *p* < 0.05 corrected for multiple comparisons, circles non-significant correlation values. Black represents the correlation during the task and red points correlation during rest. Black circles with green infill are non-significant task values.

**Figure 3 bioengineering-11-00224-f003:**
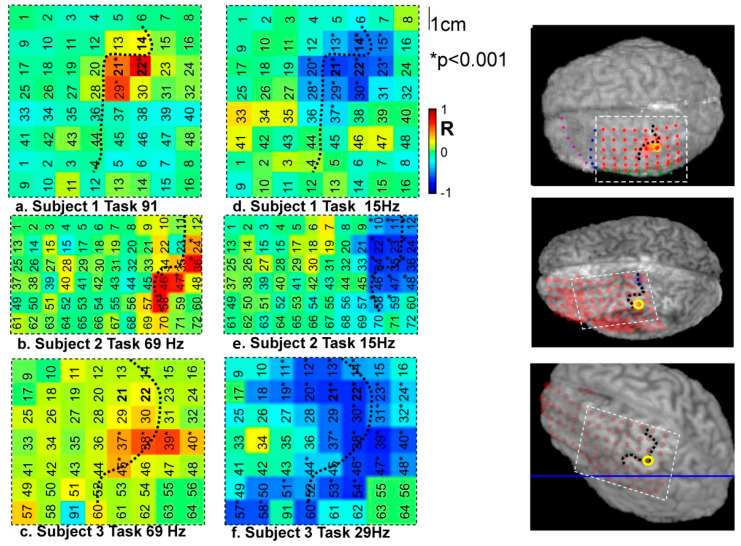
Spatial pattern of correlations between ECoG and fMRI during finger tapping.

**Figure 4 bioengineering-11-00224-f004:**
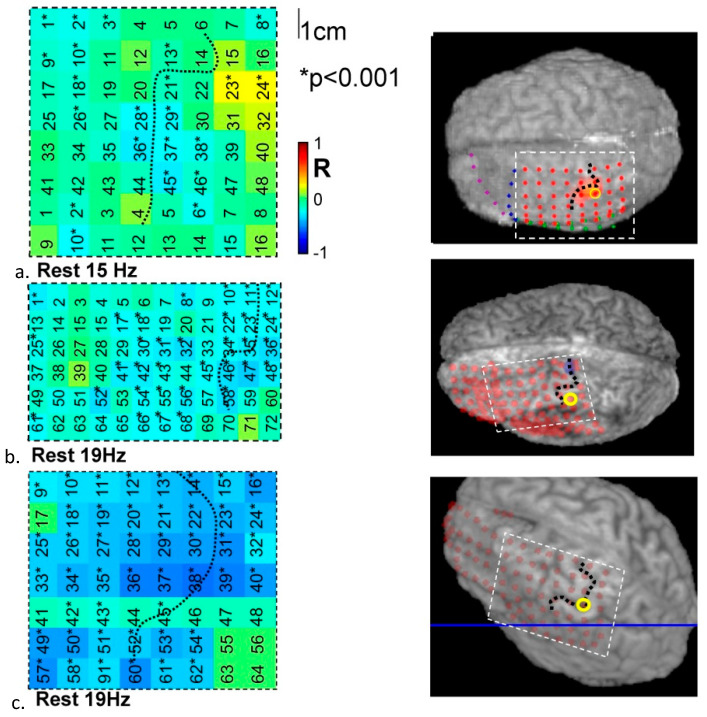
Spatial pattern of correlations between ECoG and fMRI during rest. In (**a**–**c**) the correlation for each subject was mapped spatially over the cortex for the significant negative peak in correlation (beta frequency range). Note positive correlations in the gamma range were not in general significant at rest and so were not spatially mapped.

**Figure 5 bioengineering-11-00224-f005:**
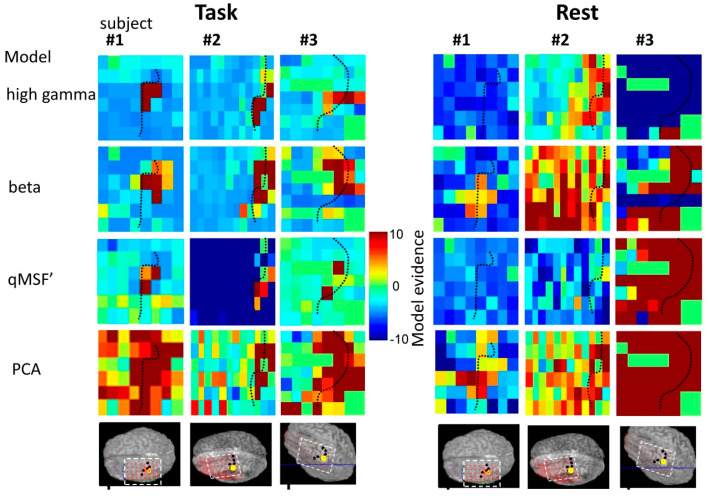
Spatial performance of different BOLD predictor models. Maps of the model evidence for a particular model and subject across the cortical grid, for the task (left hand side 4 × 3 panels) and rest (right hand side 4 × 3 panel) conditions. Each row represents a different model. Colour scale: dark red means strong evidence for the model and blue, that the model is less predictive than the null (noise only) model.

**Table 1 bioengineering-11-00224-t001:** Subject and ECoG recording information. In red are the electrodes recording the seizure onset zone (we did not record those from patient #3 during simultaneous ECoG-fMRI); in light blue are electrodes from which we recorded ECoG-fMRI in white (in addition to red electrode in patient #3) are electrodes from which we did not record ECoG-fMRI.

		Implantation Scheme
**Patient ID**	#1	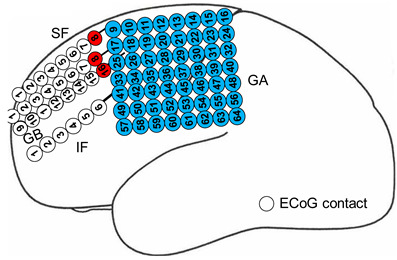
**Epilepsy**	FLE	
**Anatomical location of electrodes**	- L pre/postcentral gyrus - L supramarginal gyrus - I (IFG) and M (MFG) frontal gyri	
**Type of electrodes**	two 6-contact strips, one 8 × 8 contact grid, one 2 × 8 contact grid	
**Patient ID**	#2	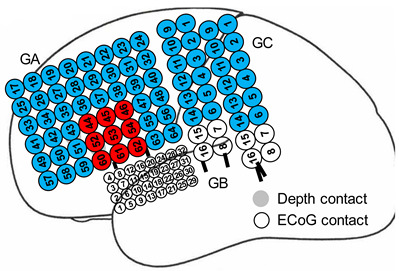
**Epilepsy**	FLE	
**Anatomical location of electrodes**	- L frontal lobe (laterally and inferiorly) - L M (MFG) and I (IFG) frontal gyri - L temporal lobe	
**Type of electrodes**	one 6 × 8 contact grid, two 2 × 8 contact grids, one 4 × 8 high-density contact grid, two 6-contact strips, two 6-contact depths	
**Patient ID**	#3	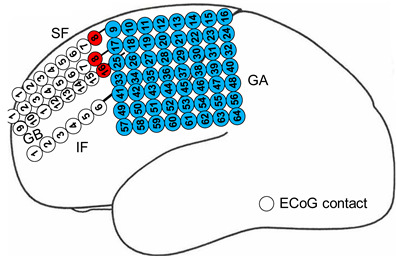
**Epilepsy**	FLE	
**Anatomical location of electrodes**	- L frontal and parietal convexity - L frontal pole - L S frontal gyrus (SFG) - L I frontal gyrus - L mesial frontal surface	
**Type of electrodes**	one 8 × 8 contact grid, one 2 × 8 contact grid, one 8-contact strip, one 6-contact strip, one high-density 4 × 8 contact grid	

**Table 2 bioengineering-11-00224-t002:** Model evaluation.

Model Evidence						
Motor Task						
Model	#Predictors	Frequency Range (Hz)				
			#1	#2	#3	Sum
**qmsf’**	1	0–100	**48.3**	**15.4**	1.5	**65.2**
qrmsf	1	0–100	15.2	−0.4	3.3	18.1
CofM	1	0–100	−1.5	−7.2	12.5	3.8
ICofM	1	0–100	1.0	−3.6	−10.6	−13.2
I40 Hz	1	0–100	1.6	−20.2	−11.1	−29.6
**PCA**	10–18	0–100	**35.1**	**23.2**	**18.3**	**76.5**
**beta**	1	13–31	**10.8**	**16.1**	**12.7**	**39.6**
gammah	1	53–99	**39.0**	**26.9**	**12.7**	**78.6**
6-band	6	0–100	−46.4	−49.0	−57.8	−153.2
Rest						
**qmsf’**	1	0–100	−5.8	−5.8	**40.7**	29.1
qrmsf	1	0–100	−5.7	0.6	**19.8**	14.7
CofM	1	0–100	−5.1	−6.6	−4.7	−16.4
ICofM	1	0–100	−7.8	−14.9	−6.5	−29.2
I40 Hz	1	0–100	−7.7	−19.1	**19.1**	−7.7
**PCA**	10–18	0–100	1.2	**7.9**	**81.6**	**90.6**
**beta**	1	13–31	−5.9	**6.2**	**55.3**	55.6
gammah	1	53–99	−7.3	**4.0**	−15.0	−18.3
6-band	6	0–100	−180.0	−73.6	−101.6	−355.1
	multiple predictor model					
	cross spectral model		**Bold text indicates significant result**			

## Data Availability

Data will be made available to interested academic parties subject to ethical considerations.

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
