# Peer review of "Measurement of the Mapping between Intracranial EEG and fMRI Recordings in the Human Brain"

_bioengineering, 2024, doi:10.3390/bioengineering11030224_

Round 1

Reviewer 1 Report

Comments and Suggestions for Authors

The article explores the intriguing relationship between human brain activity at different temporal and spatial scales, specifically usingECoG to predict functional MRI changes in the sensorimotor cortex during rest and motor performance states, contributing important insights to our understanding of the complex interplay between neural signals across different measurement modalities. Here are some concerns:

1. One limitation of the study is the small sample size, with only three participants included. This limited pool of subjects may raise concerns about the generalizability of the findings and the robustness of the observed relationships within the study.

2.The article's structure exhibits certain deficiencies. A portion of the descriptions in the results section should better be incorporated into the Methods section. This would not only enhance the overall clarity and organization of the article but also ensure that readers have a better grasp of the experimental procedures.

3.The article's layout and formatting deviate from the MDPI template.

4. The figures in the article typically consist of multiple subplots, but they lack the necessary annotations or explanations at the figure legends

5. The paper lacks a requisite conclusory summary.  Incorporating  a concluding section would enhance the overall cohesiveness and comprehensibility of the paper.

6. From a holistic perspective, the methods employed by the authors exhibit considerable versatility. However, the specific rationale for utilizing patients with epilepsy in the study is not explicitly articulated.

Author Response

The article explores the intriguing relationship between human brain activity at different temporal and spatial scales, specifically using ECoG to predict functional MRI changes in the sensorimotor cortex during rest and motor performance states, contributing important insights to our understanding of the complex interplay between neural signals across different measurement modalities. Here are some concerns:

  1. One limitation of the study is the small sample size, with only three participants included. This limited pool of subjects may raise concerns about the generalizability of the findings and the robustness of the observed relationships within the study.

>> By the very nature of these studies, large groups are extremely challenging to obtain because the EEG electrode implantation is entirely decided by clinical considerations meaning that very few patients in our surgical series have highly similar implantations. In addition, the logistic conditions imposed by the need to acquire fMRI in the usually very short time window consisting of the period between the decision to end the clinical video-icEEG recordings and the moment of surgical extraction of the electrodes are extremely limiting. In primate neuroscience studies showing reproducibility in several animals is often used (e.g. Bastos et al use n=3 animals https://www.pnas.org/doi/epdf/10.1073/pnas.1710323115) however we acknowledge this limitation clearly in the discussion.

>> Change made: sentence added to the Discussion ‘Owing to the rarity of patients with highly similar implantations and the extremely challenging logistic constraints imposed by the limited amount of time that the icEEG electrodes are in situ, but not available to record clinically useful data, it is not easily possible to study large groups. Despite a level of reproducibility across our 3 subjects we acknowledge that these numbers preclude high confidence in their generalisation.  

2.The article's structure exhibits certain deficiencies. A portion of the descriptions in the results section should better be incorporated into the Methods section. This would not only enhance the overall clarity and organization of the article but also ensure that readers have a better grasp of the experimental procedures.

>> We agree
>> Change made: We have moved the contents alluded to between the sections in the new version.

3.The article's layout and formatting deviate from the MDPI template.

>> Change made: we have moved contents between the sections in the new version to try to conform to the template.

  1. The figures in the article typically consist of multiple subplots, but they lack the necessary annotations or explanations at the figure legends

>> Change made: We have adjusted the figure legends

  1. The paper lacks a requisite conclusory summary.  Incorporating a concluding section would enhance the overall cohesiveness and comprehensibility of the paper.

>> Change made: we have moved contents between the sections in the new version to try to conform to the template.

  1. From a holistic perspective, the methods employed by the authors exhibit considerable versatility. However, the specific rationale for utilizing patients with epilepsy in the study is not explicitly articulated.

>> We have made it clearer that the only opportunity to make these recordings is in patients that have intracranial electrodes implanted for epilepsy surgery evaluation.

Reviewer 2 Report

Comments and Suggestions for Authors

This is a research result that clearly shows the relationship between temporal features and spatial features regarding brain activity. As a result of a study on three epilepsy patients, high gamma power was identified as a variable that well demonstrated the relationship between the two features. However, despite the interesting research results, there are some points that need to be improved in terms of research methods.

1. The hand sensorimotor area observed in this study is not mentioned in the introduction. What characteristics does this area have compared to other areas? A reasonable basis must be provided as to why the selection was made.

2. The specific ways in which the results of this study can be utilized are not well described. Please provide specific clinical significance of this study.

Author Response

This is a research result that clearly shows the relationship between temporal features and spatial features regarding brain activity. As a result of a study on three epilepsy patients, high gamma power was identified as a variable that well demonstrated the relationship between the two features. However, despite the interesting research results, there are some points that need to be improved in terms of research methods.

  1. The hand sensorimotor area observed in this study is not mentioned in the introduction. What characteristics does this area have compared to other areas? A reasonable basis must be provided as to why the selection was made.

>> We are dependent on the sites of electrode implantation that are chosen solely based on clinical considerations. No research-related considerations had an influence on the implantations. We have adjusted the introduction to this effect.

  1. The specific ways in which the results of this study can be utilized are not well described. Please provide specific clinical significance of this study.

>> The relationship between BOLD fMRI and intracranial EEG is important to understand because it underlies the neurobiological interpretation of fMRI results. There is not a direct clinical significance. This is now noted in Further Limitations section.

Reviewer 3 Report

Comments and Suggestions for Authors

Dear Authors,

the manuscript is the reproposal of a paper available on an open access preprint repository (bioRxiv) since December 21, 2017 (https://doi.org/10.1101/237198).

At the time, the contents presented were innovative; as it was the first report of simultaneous fMRI-ECoG recordings in humans, as reported by Haufe et al. in a paper on the same topic ("Elucidating relationships between fMRI, ECoG, and EEG through a common natural stimulus." NeuroImage 179 (2018): 79-91).

This is undoubtedly an excellent paper. However, I do not agree with the layout of the manuscript, with the Methods (page 12) exposed after Results (page 2) and Discussion (page 9).

Furthermore, since it is a 5-year-old manuscript, no recent bibliography on the topic is cited (of the 40 references cited, the most recent - two - are from 2017), such as the aforementioned Haufe et al. (2018) and Ebrahiminia et al. (Frontiers in Neuroscience, 2022).

Best regards,

Author Response

The manuscript is the reproposal of a paper available on an open access preprint repository (bioRxiv) since December 21, 2017 (https://doi.org/10.1101/237198).

At the time, the contents presented were innovative; as it was the first report of simultaneous fMRI-ECoG recordings in humans, as reported by Haufe et al. in a paper on the same topic ("Elucidating relationships between fMRI, ECoG, and EEG through a common natural stimulus." NeuroImage 179 (2018): 79-91).

This is undoubtedly an excellent paper. However, I do not agree with the layout of the manuscript, with the Methods (page 12) exposed after Results (page 2) and Discussion (page 9).

>> The layout has been edited.

Furthermore, since it is a 5-year-old manuscript, no recent bibliography on the topic is cited (of the 40 references cited, the most recent - two - are from 2017), such as the aforementioned Haufe et al. (2018) and Ebrahiminia et al. (Frontiers in Neuroscience, 2022).

>> We agree and have updated the literature including with the helpful suggestions of the reviewer.

Round 2

Reviewer 1 Report

Comments and Suggestions for Authors

The author has appropriately addressed the raised concerns and recommends the publication of the revised manuscript.

Reviewer 3 Report

Comments and Suggestions for Authors

I found that the authors accepted the suggestions I had proposed, especially in relation to updating the references.

I have nothing else to add. I had a lot of difficulty reading the corrected manuscript, with text additions in various colors, partly deleted; nor did I understand why there are formatting instructions in Chinese. But this does not concern my comments on the first version of the manuscript.

Best regards.